# Model of Shape Memory Alloy Actuator with the Usage of LSTM Neural Network

**DOI:** 10.3390/ma17133114

**Published:** 2024-06-25

**Authors:** Waldemar Rączka, Marek Sibielak

**Affiliations:** Department of Process Control, Faculty of Mechanical Engineering and Robotics, AGH University of Krakow, al. Adama Mickiewicza 30, 30-059 Kraków, Poland; sibielak@agh.edu.pl

**Keywords:** hysteresis model, neural network, SMA, Preisach model, LSTM

## Abstract

Shape Memory Alloys (SMAs) are used to design actuators, which are one of the most fascinating applications of SMA. Usually, they are on-off actuators because, in the case of continuous actuators, the nonlinearity of their characteristics is the problem. The main problem, especially in control systems in these actuators, is a hysteretic loop. There are many models of hysteresis, but from a control theory point of view, they are not helpful. This study used an artificial neural network (ANN) to model the SMA actuator hysteresis. The ANN structure and training method are presented in the paper. Data were generated from the Preisach model for training. This approach allowed for quick and controllable data generation, making experiments thoroughly planned and repeatable. The advantage and disadvantage of this approach is the lack of disturbances. The paper’s main goal is to model an SMA actuator. Additionally, it explores whether and how an ANN can describe and model the hysteresis loop. A literature review shows that ANNs are used to model hysteresis, but to a limited extent; this means that the hysteresis loop was modelled with a hysteretic element.

## 1. Introduction

Shape Memory Alloys (SMAs) are a member of the smart materials group. They are unique materials that have the ability to return to a remembered shape due to phase transformation when we add energy to a system, e.g., heat. In SMAs, we can distinguish the one-way shape memory effect, two-way shape memory effect and superelasticity; in some alloys, we can observe rubberlike behaviour. SMAs are fascinating materials that have many applications across industries due to their unique abilities. They are used to build biomedical devices such as stents and occluders in cardiology, orthodontic wires and braces in orthodontics, clamps and spacers in orthopaedics, etc. Apart from this, SMAs are also used in minimally invasive surgical tools, such as catheters and guidewires, where their ability to change shape with temperature enables precise control and manoeuvrability within the body.

SMAs are employed in eyeglass frames, allowing them to return to their original shape even after being bent or twisted. In the textile industry, SMAs are used in smart fabrics for applications such as self-tying shoelaces, shape-changing garments, and adjustable tightness in clothing. In aerospace and defence, SMAs are used in applications requiring lightweight and reliable actuation systems, such as adaptive wing structures, deployable antennae, and morphing structures for aerodynamic control. In automotive engineering, SMAs are used in various applications, including actuators for active aerodynamics, smart materials for vibration damping, and shape memory alloy springs for active suspension systems. They are used in seismic dampers for buildings and bridges to absorb and dissipate energy during earthquakes, thus enhancing structural resilience and safety. In consumer electronics, SMAs are used in mobile phone antennas and as various types of springs for energy storage or as miscellaneous actuators. They can produce large strains and forces in response to small temperature changes, making them suitable for precise control in applications such as valves, micropositioning systems, and robotics.

This last application is the most interesting from this article’s point of view. SMAs, besides unique, beneficial phenomena, unfortunately have disadvantages. The most awkward is hysteresis, which causes many problems in control systems, especially in actuators that control position. A suitable mathematical hysteresis model could solve the problem. There are many works about modelling hysteresis in SMA, but generally, it is not easy to describe because this phenomenon in SMA depends on many factors. To solve this problem, many types of models have been created. There are various models that represent SMAs from different perspectives, such as macroscopic, mesoscopic, or microscopic. The choice of the appropriate model depends on the analysis we want to perform.

Numerous studies have been conducted on the subject of SMA modelling, with Khandelwal and Buravalla delving into various models in their publication [1]. Specifically, they examine models presented by Birman [2], Paiva and Savi [3], Smith [4], and Lagoudas [5]. Their analysis focuses on continuum models in particular. Khandelwal and Buravalla have identified a specific subset of input–output black box models that are particularly valuable for macroscopic modelling of SMA phenomena. These models are highly effective for control systems and are commonly used to describe hysteresis. The most popular models of this kind include the Preisach [6,7,8] and Duhem–Madelung models [4], but they have disadvantages that could be solved by the ANN model. Thus, recently, another model of this type emerged, incorporating an ANN. In their research, Parastoo Vahdati Yektaa et al. tackled the challenge of modelling SMAs using ANN [6]. They utilised ANN and genetic algorithms to predict a permalloy’s magnetic properties and hysteresis loop based on inputs such as sample thickness, annealing temperature, holding time, and field strength. Dang X. and Tan Y. explored modelling hysteresis with ANN in their work [7]. They combined the Preisach model with a diagonal recurrent ANN to model dynamic hysteresis. Additionally, these authors used radial basis function ANNs to model hysteresis in a piezoceramic actuator [8]. In [9], Sixdenier et al. used a feed-forward ANN to model scalar hysteresis in a rate-independent manner. Li Y. et al. employed a back propagation ANN to combine Jiles–Atherton and Preisach models for hysteresis modelling. Similarly, Nguyen Trong Tai et al. utilised a functional link ANN with a hysteresis operator to model an SMA actuator in [10], an analogical solution one can find in [11]. In [12], the authors used a relatively small ANN for the rate-dependent model of hysteresis. However, the above papers present various models that combine ANN with a model of hysteresis formulated in other ways, as we see in [13,14].

So, the article’s main goal is to adopt only an ANN for hysteresis modelling. This article presents a neural model that effectively describes the SMA actuator, focusing on addressing the complex hysteresis loop.

## 2. Materials and Methods

This article discusses the topic of SMA modelling using a neural network. It was assumed that the modelled object is an SMA actuator whose input is temperature and output is displacement; no other signals are considered in this model. The actuator is understood as an executive element that generates displacement under the influence of the supplied thermal energy. This actuator uses the one-way shape memory effect. It performs useful movement under the influence of the supplied thermal energy, while the return movement is carried out using dead mass. This highly simplified model aims to check the possibility of modelling the SMA actuator using a neural network without the classic hysteron model. As shown in the literature review, existing models of neural SMA do not model hysteresis itself. In neural models, elementary hysteresis models are used to model hysteresis.

This article used an LSTM network to model the SMA actuator, which was trained using data obtained from the Preisach model. Various data sources can be used to train the network. Since this article only deals with checking the method, a fully controllable data source was chosen. Thanks to this, you can test the method using different signals in training and testing in a quick and cheap way.

This article presents a solution that employs long short-term memory (LSTM) [15], a recurrent ANN type. LSTM is particularly suitable for describing hysteresis loops, given its capacity to memorise sequences. Additionally, LSTM, as described by Equation (1), can retain its state between predictions, making it a valuable tool for time series analysis, even when the complete time series is unavailable or when long-term predictions are required. As described in [15], LSTM is described below:(1)$$\begin{array}{c} f_{t}=\sigma_{g}\left(W_{f}x_{t}+U_{f}h_{t-1}+b_{f}\right)\\ i_{t}=\sigma_{g}\left(W_{i}x_{t}+U_{i}h_{t-1}+b_{i}\right)\\ o_{t}=\sigma_{g}\left(W_{o}x_{t}+U_{o}h_{t-1}+b_{o}\right)\\ {\tilde{c}}_{t}=\sigma_{c}\left(W_{c}x_{t}+U_{c}h_{t-1}+b_{c}\right)\\ c_{t}=f_{t}⨀c_{t-1}+i_{t}⨀{\tilde{c}}_{t}\\ h_{t}=o_{t}⨀\sigma_{t}\left(c_{t}\right) \end{array}$$
where:




$f_{t}$

forget gate’s activation vector;

$i_{t}$

input gate’s activation vector;

$o_{t}$

output gate’s activation vector;

${\tilde{c}}_{t}$

cell input activation vector;

$c_{t}$

cell state vector;

$h_{t}$

hidden state vector;

$x_{t}$

input vector to ANN;

$\sigma_{g}$

sigmoid function;

$\sigma_{c}$

hyperbolic tangent function;

$b_{*}$

bias vector subscript * means the same as above;

$W_{*},U_{*}$

weight matrices of input and connections, respectively, subscript * means input gate *i*, output gate *o*, forget gate *f*, memory cell *c*;⨀denotes the Hadamard product.


An ANN with the structure shown below consists of a course input layer with sequence data inputs and an output layer. Between these two layers, there are one or more LSTM layers with a number of hidden units, a fully connected layer and a regression layer as a single-output layer. During experiments, there are changed numbers of LSTM layers, hidden units, and training epochs. The network was trained based on data from the discrete Preisach hysteresis model. A number of hidden units in the LSTM layer were changed during each experiment.

The ANN was trained using data obtained from the Preisach model. This was because the data from the mathematical model are always fully controllable, and data are repeatable. We can easily prepare data sets exactly as we want. Since this article focuses on testing the possibility of training ANN hysteresis, the data source was not significant.

Preisach model of hysteresis is the result of an infinite number of elementary hysteresis operators named hysterons, shown in Figure 1. Each hysteron functions like a relay that switches on when the input signal u(t) reaches the value α from left side as it follows the abcde curve in Figure 1, so in case u(t)≥ α, ${\overset{ˇ}{\gamma }}_{\alpha \beta }\left(u\left(t\right)\right)=+1$. Conversely, the hysteron switches off, and the hysteresis operator takes the value ${\overset{ˇ}{\gamma }}_{\alpha \beta }\left(u\left(t\right)\right)=-1$, when the input u(t) decreases along the edfba curve (u(t)≤ β). The hysteron is defined by two independent parameters, α and β, and it satisfies the condition α≥ β.

The Preisach hysteresis model is formulated below.
(2)$$f\left(t\right)=\iint_{\alpha \geq \beta }\mu \left(\alpha ,\beta \right){\overset{ˇ}{\gamma }}_{\alpha \beta }\left(u\left(t\right)\right)d\alpha d\beta$$
where:




μ(α,β)

weight function;

u(t)

input signal;

${\overset{ˇ}{\gamma }}_{\alpha \beta }$

hysteron.


To obtain data for training and testing, the Preisach model was excited by sine and triangle waves with various amplitudes; for each amplitude, multiple periods per amplitude were prepared. During each experiment, the data used to train ANN were the same. The data used to testify and to train networks were prepared separately. The model’s response, as well as the excitation signal, were used as data to train the network. Separate data were generated to test the network at other amplitudes.

The Preisach model was used to obtain data for training and testing. A test data set consisting of 4 sequences of numbers generated as time sequences of various signals was prepared. Each of them contained 13 different amplitudes, thanks to which the temperatures varied from 0 °C to 40 °C, 45 °C, 55 °C, 60 °C, 62 °C, 64 °C, 66 °C, 68 °C, 70 °C, 72 °C, 74 °C, 78 °C, and 81 °C. This non-linear distribution was created to best match these waveforms to the modelled hysteresis, the loop of which is in the range of 38–78 °C. These sequences were organized in such a way that two of them were series of triangular waveforms with amplitudes varying as indicated above, with the second of the series having a frequency ten times lower. The next two excitation waveforms were sinusoidal and were organized similarly; the second in the series had a ten times lower frequency. With this signal, the Preisach model was activated. The results obtained from the Preisach model were compared with the test signal (described above), which constituted the training signal. The length of this training signal was 15,736 points. During each experiment, the data used to train ANN were the same.

The data used to testify trained networks were prepared separately. The test signal was organized in the same way as the training signal, but it was shortened to two triangular waveforms with amplitudes such that the temperature varied from 0 °C to 58 °C, 61 °C, 65 °C, 67 °C, 71 °C, 75 °C, and 77 °C.

The data obtained from the Preisach model were used to train and test a network with the structure mentioned earlier. During testing, the number of hidden units in the LSTM layer was modified, along with the number of learning epochs. After training the network, it was tested with a previously prepared test signal, and the results were presented in the form of charts, which showed the relationship between the shortening of the SMA wire and its temperature. The SMA wire was the actuator tested, and the input was the temperature in Celsius degrees. The percentage shortening of the SMA wire was given in numerical form.

The motivation for conducting this research was a desire to answer the following questions: is it possible to model hysteresis with only ANN without a hysteretic element? What number of hidden units should we use to obtain an exact model, and what should the structure of ANN be? A number of tests were conducted to answer these questions. The chosen results are shown in this article.

At the beginning, a series of tests aimed to determine the number of training epochs. To achieve this, the same dataset was used to train a network with varying numbers of hidden units and training epochs while keeping all other parameters the same. The training sets and test signals also remained unchanged. The experiment was conducted for the following numbers of hidden units: 10, 50, 100, 300, 500, 1000 and 100, 200, 300, 500, 1000 epochs. All results were obtained under the same conditions. After analysing the results of these experiments, 1000 training epochs were chosen. Next, experiments were prepared and conducted with different numbers of LSTM layers. The structure of the network is shown below. As one can see, it consists of an input layer, one or more LSTM layers, a fully connected layer and an output layer. The number of LSTM layers was changed between each series of experiments. Four series of experiments were conducted with one, two, three, and five LSTM layers.
*Networklayers = [sequenceInputLayer(featureDimension) …*Input layer*lstmLayer(numHiddenUnits,”OutputMode”,”sequence”) …*1st LSTM layer*lstmLayer(numHiddenUnits,”OutputMode”,”sequence”) …*2nd LSTM layer*lstmLayer(numHiddenUnits,”OutputMode”,”sequence”) …*3rd LSTM layer*lstmLayer(numHiddenUnits,”OutputMode”,”sequence”) …*4th LSTM layer*lstmLayer(numHiddenUnits) …*5th LSTM layer*fullyConnectedLayer(numResponses) …*fully connected layer*regressionLayer];*Output layer

Other parameters of the training environment are shown below.


*solverName = ‘adam’;*



*options = trainingOptions(solverName, …*



*‘MaxEpochs’, maxEpochs, …*



*‘MiniBatchSize’, miniBatchSize, …*



*‘GradientThreshold’, 10, …*



*‘Plots’,‘training-progress’, …*



*‘ExecutionEnvironment’, ‘gpu’, …*



*‘TrainRateDropPeriod’, 100, …*



*‘Verbose’,0);*


## 3. Results

The figures below provide a summary of data obtained from a number series of tests for ANN with 1, 2, 3, 5 LSTM layers. The results of the experiments are displayed in Figure 2, Figure 3, Figure 4, Figure 5, Figure 6, Figure 7, Figure 8, Figure 9, Figure 10, Figure 11, Figure 12 and Figure 13. The results are chosen to show the behaviour of ANN and to show how structure influences results. The results are organised in such a manner as to make the analysis of them easier. Figures are sorted from the simplest structure of ANN to the most complicated. Each series conducted tests with different numbers of hidden units. In each series, hidden units were set to 10, 50, 100, 300, 500, and 1000. In each series of tests, a number of hidden units were changed in the same manner. For the purity of the tests, no other parameters of the ANN itself were changed, and no training parameters were changed. The results of the tests for each series of tests are shown below. In Figure 2, Figure 3 and Figure 4, ANN has only one LSTM layer; in Figure 5, Figure 6 and Figure 7, ANN has two LSTM layers, while in Figure 8, Figure 9 and Figure 10, ANN has three LSTM layers, and at the end, in Figure 11, Figure 12 and Figure 13, ANN has five LSTM layers.

Each figure shows the reference response obtained from the Preisach model drawn with a solid black line and the response of the trained ANN drawn with a solid red line. Each of the figures includes basic information about a number of hidden units and a number of training epochs. The arrows show the branches responding to increasing (up arrow) and decreasing (down arrow) input signals. The response is a relative displacement understood as the percentage shortening of the SMA length of the wire in response to a change in wire temperature.

Figure 2, Figure 3 and Figure 4 show the test signal responses of an ANN containing one LSTM layer. As we can see, the answers differ significantly from the pattern. In fact, it should be said that this model does not describe the hysteresis of the SMA wire. Although the model differs qualitatively nor quantitatively from the model, it can be seen that a hysteresis loop exists and that it responds to a change in the amplitude of the forcing signal.

In turn, Figure 5, Figure 6 and Figure 7 show the responses to the test signal of an ANN containing two LSTM layers. As we can see, the responses of this network describe the hysteresis of the SMA wire much better. For a network composed of two LSTM layers containing 500 and 1000 hidden units, the network responses are satisfactory in the main part. Even for small temperature values, the model has large errors. If we use a model built and trained in this way, we can obtain satisfactory answers for temperatures around the actuator operating range. This model could be sufficient in many cases.

Figure 8, Figure 9 and Figure 10 show the test signal responses of an ANN containing three LSTM layers. In this case, the responses shown in Figure 9b and Figure 10 are almost perfect in the actuator’s operating range, i.e., in the temperature range from 20 °C to 80 °C, but the model is incorrect for lower temperatures.

The final test series shown in Figure 11, Figure 12 and Figure 13 shows the responses to the test signal of an ANN containing five LSTM layers. In this case, the responses shown in Figure 12 and Figure 13 are proper over the entire range, although not perfect like in Figure 9b and Figure 10 with the operating range narrowed to temperatures from the range of 20 °C to 80 °C. In this case, the responses of the network containing 10 and 50 hidden units are wrong.

To sum up, it is possible to find an ANN with such a number of LSTM layers and hidden units that the hysteresis of the SMA actuator is described in the same way as the Preisach model. However, when using ANN, we can expect that it will also describe various types of artefacts observed in the characteristics of real SMA actuators obtained in a laboratory. What the Preisach model cannot take into account, ANN can learn. The selection of the ANN structure and parameters will be time-consuming, but such a model may be more perfect than the currently used hysteresis models.

## 4. Conclusions

In this paper, an ANN was utilised to model an SMA actuator, with the addition of an actuator exhibiting hysteresis. A literature analysis revealed that while there are existing neural models for hysteresis objects, the hysteresis loop itself has not yet been modelled using ANN but by other mathematical models of elementary hysteresis. Thus, the main goal of the article was to capture the hysteresis phenomenon using ANN, and the results demonstrated that the LSTM network is effective in doing so. The second goal of the article was to find how to design the ANN. It is worth knowing how to design an ANN, and what structure and what parameters to use to obtain fully satisfactory results. So, the results for four different structures of ANN and for different numbers of hidden units are shown in the article. The results show the way in which we can find the optimal structure and parameters of ANN. This article does not answer what optimal ANNs are, but it shows where the solution is lying and what the possibilities are. To answer what the structure and parameters of ANN should be, we need to answer a few questions, e.g., what precision we need, whether we need to describe specific artefacts or not, what time of calculations is satisfactory, etc.

We see that we can find an optimal ANN. Now, the more interesting question is if the ANN could change to describe changes in SMA features, and if it can, what limitations and risks are hidden in the adaptation mechanism and how can we design an ANN to ensure the potentiality of the network to satisfy the needs of the adaptation mechanism.

## Figures and Tables

**Figure 1 materials-17-03114-f001:**
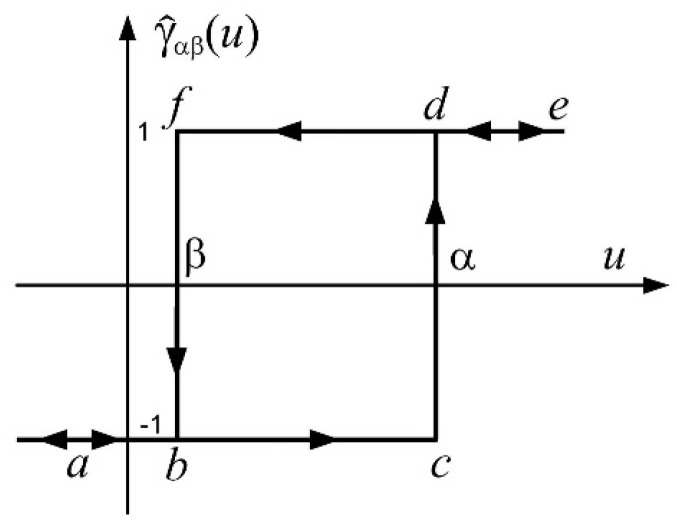
Elementary hysteresis operator—hysteron.

**Figure 2 materials-17-03114-f002:**
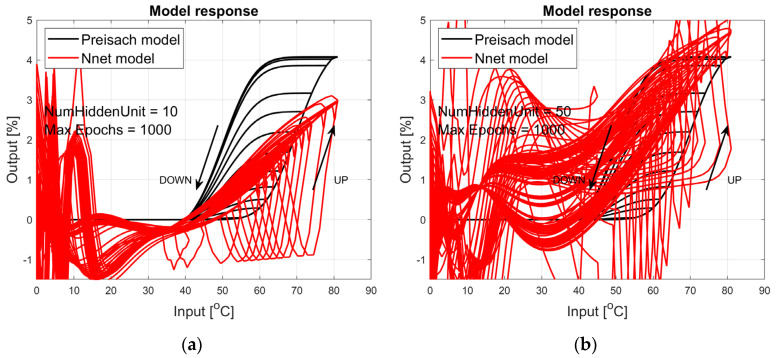
Plot of relative displacement as a function of temperature. The response of the Preisach model is the black line. The response of the LSTM model with one LSTM layer is in the red line.

**Figure 3 materials-17-03114-f003:**
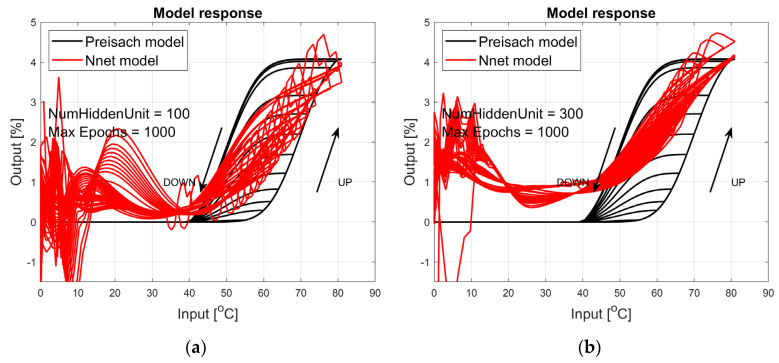
Plot of relative displacement as a function of temperature. The response of the Preisach model is the black line. The response of the LSTM model with one LSTM layer is in the red line.

**Figure 4 materials-17-03114-f004:**
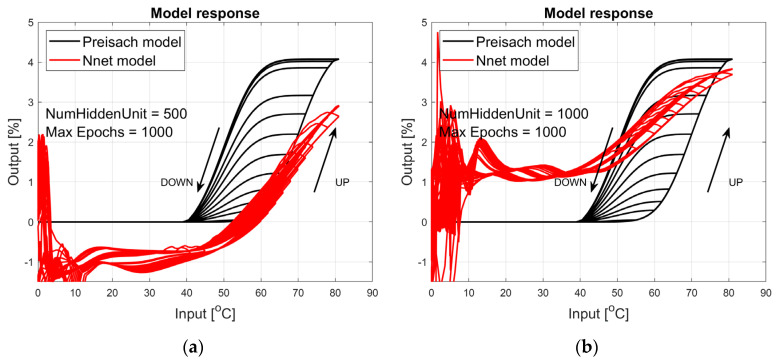
Plot of relative displacement as a function of temperature. The response of the Preisach model is the black line. The response of the LSTM model with one LSTM layer is in the red line.

**Figure 5 materials-17-03114-f005:**
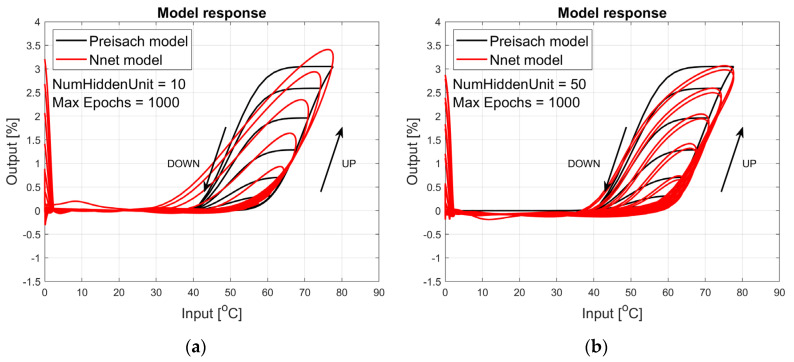
Plot of relative displacement as a function of temperature. The response of the Preisach model is the black line. The response of the LSTM model with two LSTM layers is in the red line.

**Figure 6 materials-17-03114-f006:**
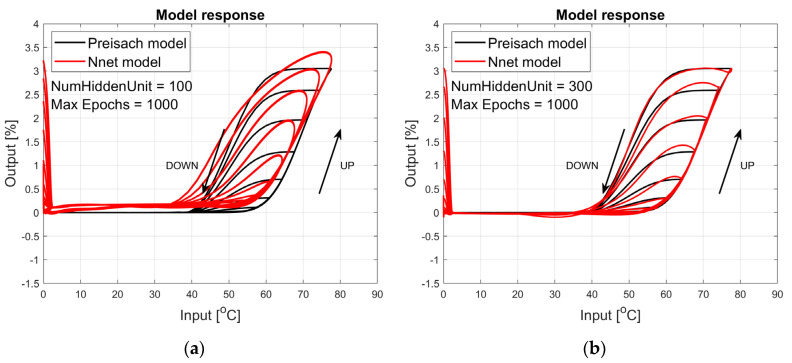
Plot of relative displacement as a function of temperature. The response of the Preisach model is the black line. The response of the LSTM model with two LSTM layers is in the red line.

**Figure 7 materials-17-03114-f007:**
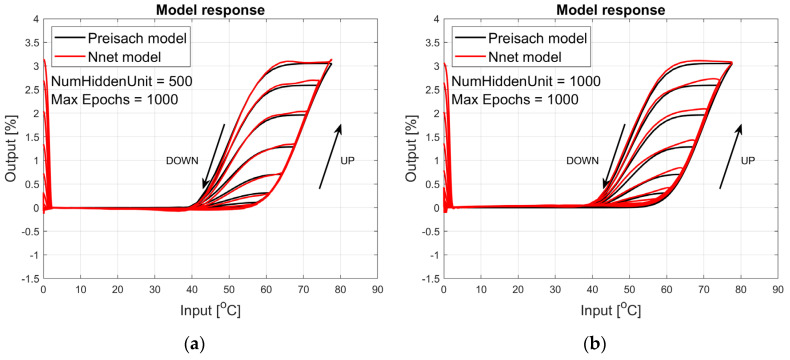
Plot of relative displacement as a function of temperature. The response of the Preisach model is the black line. The response of the LSTM model with two LSTM layers is in the red line.

**Figure 8 materials-17-03114-f008:**
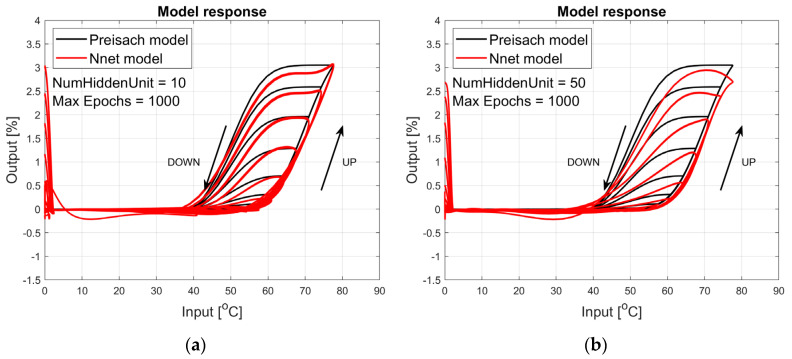
Plot of relative displacement as a function of temperature. The response of the Preisach model is the black line. The response of the LSTM model with three LSTM layers is in the red line.

**Figure 9 materials-17-03114-f009:**
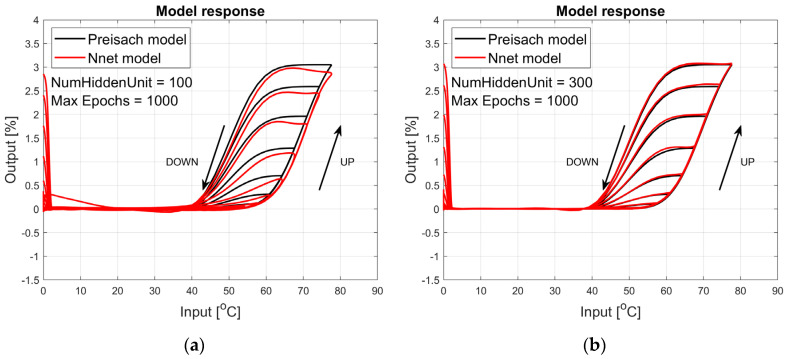
Plot of relative displacement as a function of temperature. The response of the Preisach model is the black line. The response of the LSTM model with three LSTM layers is in the red line.

**Figure 10 materials-17-03114-f010:**
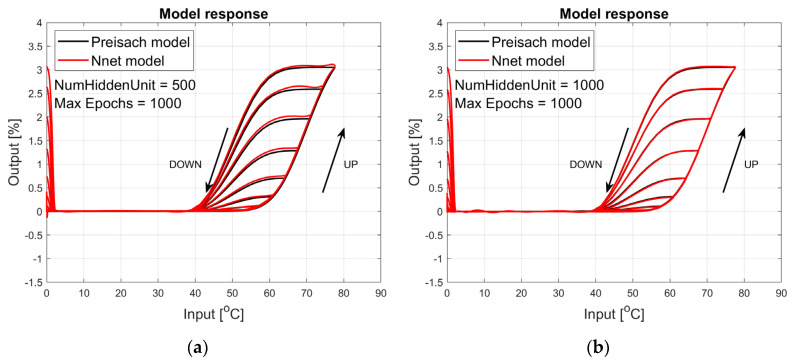
Plot of relative displacement as a function of temperature. The response of the Preisach model is the black line. The response of the LSTM model with three LSTM layers is in the red line.

**Figure 11 materials-17-03114-f011:**
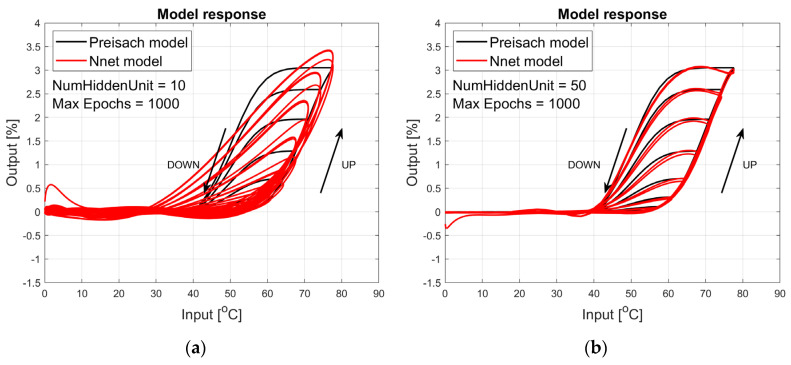
Plot of relative displacement as a function of temperature. The response of the Preisach model is the black line. The response of the LSTM model with five LSTM layers is in the red line.

**Figure 12 materials-17-03114-f012:**
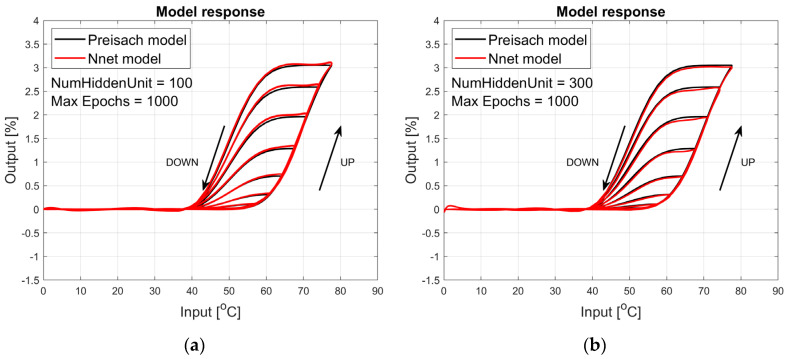
Plot of relative displacement as a function of temperature. The response of the Preisach model is the black line. The response of the LSTM model with five LSTM layers is in the red line.

**Figure 13 materials-17-03114-f013:**
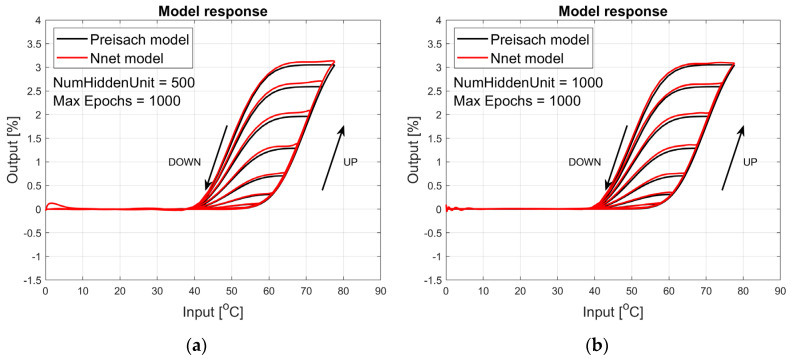
Plot of relative displacement as a function of temperature. The response of the Preisach model is the black line. The response of the LSTM model with five LSTM layers is in the red line.

## Data Availability

Data are contained within the article.

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
