# Peer review of "Model of Shape Memory Alloy Actuator with the Usage of LSTM Neural Network"

_materials, 2024, doi:10.3390/ma17133114_

Round 1

Reviewer 1 Report

Comments and Suggestions for Authors

Reviewer’s comment

The article ntitled ‘(Model of shape memory alloy actuator with the usage of LSTM neural network )’ has been submitted for publication in the (Journal of Materials.)

The research focuses on studying the prediction of hysteresis models in the relationship between Shape Memory Alloys and temperature changes using artificial neural networks (ANN). To identify significant factors in ANN modeling structures, comparisons were made based on the number of long short-term memory (LSTM) units, hidden layers, number of epochs, and the intended use of the Preisach model.

1.     In all research explanations, authors should avoid repetitive word usage and refrain from using colloquial terms such as "of course," opting for more formal or academic language instead.

2.     In the materials and methods section, the author should provide a detailed explanation of the experimental design, including the factors considered in the study, presented in a table format.

3.     In the results section, paragraph 3, the authors should elaborate more on the reasons or mechanisms behind why the results of the ANN model cannot accurately describe the hysteresis phenomenon.

4.     In the results section, paragraph 4, where better results are mentioned, authors should provide a more detailed explanation of the factors contributing to the improved results and specify the specific static value of the large errors observed in the low-temperature results.

5.     In the results section, paragraph 5, authors should explain the significance of the perfect results described in Figures 9b and 10 in terms of static theory and specify the specific static value of the large errors observed in the low-temperature results.

6.     In the results section, paragraph 6, where it is mentioned "although not perfect like in Figures 9b and 10," authors should explain the implications within the context of static theory and provide supporting reasons for the results.

7.     In the plot of relative results, authors should estimate the significant factors in Figures 8a and 11a, considering the differences in results when the number of hidden units and epochs remained the same while LSTM was increased.

8.     In the conclusion section, when describing the effectiveness of the LSTM network, authors should provide statistical evidence supporting this assertion.

9.     In the conclusion section, authors should provide statistical evidence supporting the claim that the ANN solution demonstrates the lying part and offers possibilities, as mentioned.

Comments on the Quality of English Language

Moderate editing of English language required

Author Response

Dear Sirs

Answers are included in attached file. 

Reviewer 2 Report

Comments and Suggestions for Authors

This paper used an artificial neural network to model the SMA actuator hysteresis. Several critical aspects require further clarification and elaboration.

1.       Details of Input and Output Variables:

Please provide a comprehensive discussion of the input and output variables used in the model. This should include the data format, such as the length of the input sequence data.

Describe the data pre-processing steps in detail. Specify whether data normalization or standardization techniques were applied.

2.       Training and Testing Dataset Information:

Specify the sizes of the training and testing datasets, including the number of labeled samples in each.

Explain the preparation process for these datasets. How were the datasets split and organized for training and testing purposes?

3.       Figure 2 – 13 compared the relative displacement as a function of temperature derived from Preisach model and LSTM model. The results indicate better agreement between the two models as the number of LSTM layers and neurons increases. Clarify whether these results pertain to the training phase or the testing phase. Ensure that the results presented are from the testing phase, as using training data for validation can lead to misleading conclusions.

4.       Potential Overfitting Concerns:

The model's configuration, with 5 layers and 1000 neurons, suggests a high number of weights, raising concerns about potential overfitting.

Overfitting occurs when the model performs well on training data but poorly on testing data. To address this, plot and compare the loss functions for both the training and testing phases. Assess and discuss the model’s performance metrics in both phases to ensure a balanced evaluation.

Comments on the Quality of English Language

1.       Line 7: ‘SMA’ needs a full description ‘Shape memory alloy (SMA)’

2.       Line 15-16: ‘The paper's main goal is to model an SMA actuator, but it was also interesting to see if an ANN could describe the hysteresis loop and how’ – ‘The paper's main goal is to model an SMA actuator. Additionally, it explores whether and how an ANN can describe and model the hysteresis loop’

3.       Line 158: ‘Of course, the data used to testify to trained networks were prepared separately.’ – ‘Of course, the data used to testify to train networks were prepared separately.’

Author Response

(The authors gave the same response as above.)

Reviewer 3 Report

Comments and Suggestions for Authors

The article is devoted to modern topics and is related to modeling the properties of shape memory alloys.

The article contains textual and semantic coincidences with the work -

10.1109/ICCC57093.2023.10178984

It is necessary to reduce the percentage of repetition, and also explain how the new and old work are fundamentally different.

In the introduction, it is necessary to reveal in more detail the essence of the shape memory effect.

Since the title of the article contains the word “actuator,” it is necessary to describe its operating principle.

In the experiment, it is necessary to indicate the chemical composition of the shape memory alloy. (Titanium nickelide? 50.5 at% Ni).

Some graphs can be combined into one figure, since the analysis of the results is done by enumeration.

There are very few references to real experiments of other researchers with shape memory alloys

Not all references are formatted according to journal standards.

Author Response

(The authors gave the same response as above.)

Round 2

Reviewer 1 Report

Comments and Suggestions for Authors

The quality of paper has been improved.

The paper has been revised accordingly to reviewer's comment.

It can be accepted for publication. 

Reviewer 2 Report

Comments and Suggestions for Authors

The manuscript presents a well-structured and thoroughly researched study. The paper meets the standards of quality and rigor required for publication and would be a valuable addition to the journal.

Reviewer 3 Report

Comments and Suggestions for Authors

The reviewer does not quite agree that some of the points mentioned in the comments are not important in this work. However, it should be noted that a lot of work has been done to improve the article and it can be published